# Enhancing Emotion Recognition in Conversation via Multi-view Feature Alignment and Memorization

**Guiyang Hou, Yongliang Shen, Wenqi Zhang, Wei Xue, Weiming Lu***

College of Computer Science and Technology , Zhejiang University

{gyhou, syl, zhangwenqi, lokilanka, luwm}@zju.edu.cn

## Abstract

Emotion recognition in conversation (ERC) has attracted increasing attention in natural language processing community. Previous work commonly first extract semantic-view features via fine-tuning pre-trained language models (PLMs), then models context-view features based on the obtained semantic-view features by various graph neural networks. However, it is difficult to fully model interaction between utterances simply through a graph neural network and the features at semantic-view and context-view are not well aligned. Moreover, the previous parametric learning paradigm struggle to learn the patterns of tail class given fewer instances. To this end, we treat the pre-trained conversation model as a prior knowledge base and from which we elicit correlations between utterances by a probing procedure. And we adopt supervised contrastive learning to align semantic-view and context-view features, these two views of features work together in a complementary manner, contributing to ERC from distinct perspectives. Meanwhile, we propose a new semi-parametric paradigm of inferencing through memorization to solve the recognition problem of tail class samples. We consistently achieve state-of-the-art results on four widely used benchmarks. Extensive experiments demonstrate the effectiveness of our proposed multi-view feature alignment and memorization[1].

## 1 Introduction

Emotional intelligence is an advanced capability of conversational AI systems. A fundamental and critical task in this domain is emotion recognition in conversation (ERC), which aims to identify the emotions conveyed in each utterance within the dialogue context (Poria et al., 2019b).

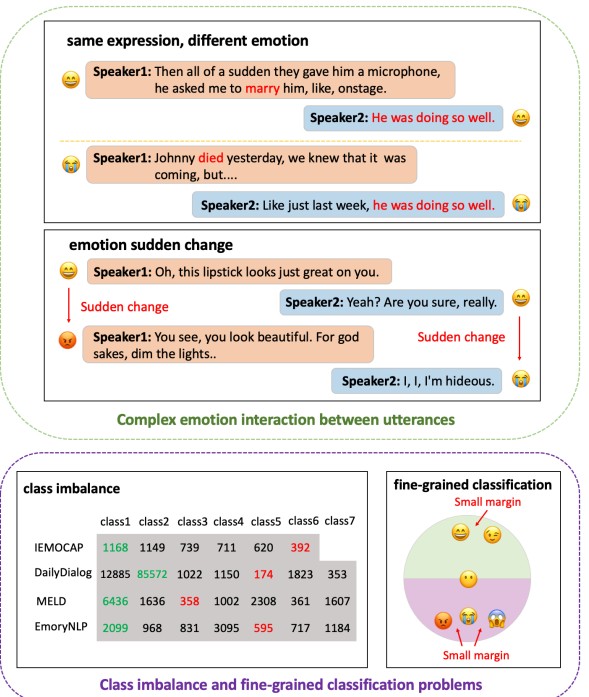

Figure 1: Major challenges in the ERC task.

Unlike the basic emotion classification (EC) task (Yin and Shang, 2022), ERC is a more practical endeavor that involves predicting the emotion label of each utterance based on the surrounding context. Previous methods (Ghosal et al., 2019; Ishiwatari et al., 2020; Shen et al., 2021b) commonly follow a two-step paradigm of first extracting semantic-view features via fine-tuning PLMs and then modeling context-view features based on the obtained semantic-view features by various graph neural networks, such as GCN (Kipf and Welling, 2016), RGCN (Ishiwatari et al., 2020), GAT (Veličković et al., 2017). Considering the complexity of emotions, some recent works (Song et al., 2022; Hu et al., 2023) use supervised contrast loss as an auxiliary loss to improve the feature discriminability between different classes of samples. Although these methods have achieved excellent performance on ERC task, three issues still remain: (1) As illus-

---

*Corresponding author.

[1] https://github.com/gyhou123/MFAM.

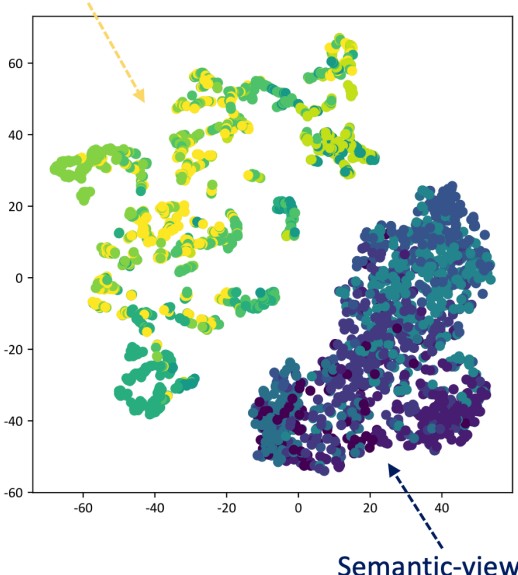

Figure 2: T-SNE (Van der Maaten and Hinton, 2008) visualization of features at semantic-view and context-view.

trated in Figure 1, the interaction between utterances in a conversation is very complex, and it's difficult to fully model this interaction simply through a graph neural network. (2) Both semantic-view and context-view are important for ERC. The former focuses on the emotion conveyed by the independent utterance, while the latter offers clues about the emotional context. These two views of information work together in a complementary manner, contributing to ERC from distinct perspectives. However, as illustrated in Figure 2, semantic-view and context-view are not well aligned. (3) Due to the smaller number of samples in the tail class, it is difficult for the parametric model to learn the pattern of the tail class during training. How to effectively recognize samples of the tail class remains an issue to be solved.

To address these issues, we propose **M**ulti-view **F**eature **A**lignment and **M**emorization (**MFAM**) for ERC. Firstly, we treat the pre-trained conversation model (PCM) (Gu et al., 2021) as a prior knowledge base and from which we elicit correlations between utterances by a probing procedure. The correlation between utterances will be used to form the weights of edges in the graph neural network and participate in the interactions between utterances. Secondly, we adopt supervised contrastive learning (SCL) to align the features at semantic-view and context-view and distinguish semantically similar emotion categories. Unlike the regular SCL,

both semantic-view and context-view features will participate in the computations of SCL. Finally, we propose a new semi-parametric paradigm of inferencing through memorization to solve the recognition problem of tail class samples. We construct two knowledge stores, one from semantic-view and the other from context-view. Semantic-view knowledge store regarding semantic-view features and corresponding emotion labels as memorized key-value pairs, and the context-view knowledge store is constructed in the same way. During inference, our model not only infers emotion through the weights of trained model but also assists decision-making by retrieving examples that are memorized in the two knowledge stores. It's worth noting that semantic-view and context-view features have been well aligned, which will facilitate the joint retrieval of the semantic-view and context-view.

The main contributions of this work are summarized as follows: (1) We propose multi-view feature alignment for ERC, which aligns semantic-view and context-view features, these two views of features work together in a complementary manner, contributing to ERC from distinct perspectives. (2) We propose a new semi-parametric paradigm of inferencing through memorization to solve the recognition problem of tail class samples. (3) We treat the PCM as a prior knowledge base and from which we elicit correlations between utterances by a probing procedure. (4) We achieve state-of-the-art results on four widely used benchmarks, and extensive experiments demonstrate the effectiveness of our proposed multi-view feature alignment and memorization.

## 2   Related Work

### 2.1   Emotion Recognition in Conversation

Most existing works (Ghosal et al., 2019; Ishiwatari et al., 2020; Shen et al., 2021b) commonly first extract semantic-view features via fine-tuning PLMs and then model context-view features based on the obtained semantic-view features by various graph neural networks, there are also some works (Zhong et al., 2019; Shen et al., 2021a; Majumder et al., 2019) that use transformer-based and recurrence-based methods to model context-view features. It's worth noting that self-attention (Vaswani et al., 2017) in transformer-based methods can be viewed as a fully-connected graph in some sense. Some recent works (Li et al., 2021; Hu et al., 2023)use supervised contrastive loss as an auxiliary loss to

enhance feature discriminability between samples of different classes. In the following paragraphs, we provide a detailed description of graph-based methods and methods using supervised contrastive loss.

DialogueGCN(Hu et al., 2021) treats the dialogue as a directed graph, where each utterance is connected with the surrounding utterances. DAG-ERC(Shen et al., 2021b) uses a directed acyclic graph to model the dialogue, where each utterance only receives information from past utterances. CoG-BART(Li et al., 2021) adapts supervised contrastive learning to make different emotions mutually exclusive to identify similar emotions better. SACL(Hu et al., 2023) propose a supervised adversarial contrastive learning framework for learning generalized and robust emotional representations.

## 2.2 Memorization

Memorization-based methods (or non/semi-parametric methods) performs well under low-resource scenarios, and have been applied to various NLP tasks such as language modeling(Khandelwal et al., 2019), question answering(Kassner and Schütze, 2020), knowledge graph embedding(Zhang et al., 2022) and relation extraction(Chen et al., 2022).

## 2.3 Probing PLMs

Some work has shown that PLMs such as BERT(Devlin et al., 2018), RoBERTa(Liu et al., 2019), ELECTRA(Clark et al., 2020) store rich knowledge. Based on this, PROTON(Wang et al., 2022) elicit relational structures for schema linking from PLMs through a unsupervised probing procedure. Unlike PROTON operating at the word level, our probe procedure is based on PCM and operates at the utterance level.

## 3 Methodology

We introduce the definition of ERC task in section 3.1, and from section 3.2 to section 3.5, we introduce the proposed MFAM in this paper. The overall framework of MFAM is shown in Figure 3.

## 3.1 Definition

Given a collection of all speakers $\mathcal{S}$, an emotion label set $\mathcal{Y}$ and a conversation $\mathcal{C}$, our goal is to identify speaker's emotion label at each conversation turn. A conversation is denoted as $[(s_1, u_1), (s_2, u_2), \cdots, (s_N, u_N)]$, where $s_i \in \mathcal{S}$

is the speaker and $u_i$ is the utterance of $i$-th turn. For utterance $u_i$, it is comprised of $n_i$ tokens $u_i = [\omega_{i,1}, \omega_{i,2}, \cdots, \omega_{i,n_i}]$.

## 3.2 Multi-view Feature Extraction

In this section, we will introduce how to extract the semantic-view and context-view features of utterance.

### 3.2.1 Semantic-view Feature Extraction

**Semantic Feature Extraction**   We employ PLM to extract the semantic feature of utterance $u_i$. Following the convention, the PLM is firstly fine-tuned on each ERC dataset, and its parameters are then frozen while training. Following Ghosal et al. (2020), we employ RoBERTa-Large (Liu et al., 2019) as our feature extractor. More specifically, for each utterance $u_i$, we prepend a special token $[CLS]$ to its tokens, making the input a form of $\{[CLS], w_{i,1}, w_{i,2}, \cdots, w_{i,n_i}\}$. Then, we average the $[CLS]$'s embedding in the last 4 layers as $u_i$'s semantic feature vector $x_i \in \mathbb{R}^{d_u}$.

**Commensense Feature Extraction**   We extract six types ($xIntent$, $xAttr$, $xNeed$, $xWant$, $xEffect$, $xReact$) of commensense feature vectors related to the utterance $u_i$ from COMET(Bosselut et al., 2019):

$$c_i^j = COMET(w_{i,1}, w_{i,2}, \cdots, w_{i,n_i}, r_j) \quad (1)$$

where $r_j(1 \leq j \leq 6)$ is the token of $j$-th relation type, $c_i^j \in \mathbb{R}^{d_c}$ represent the feature vectors of the $j$-th commensense type for $u_i$.

Semantic-view feature is obtained by concatenate semantic feature vectors with their corresponding six commensense feature vectors as follows:

$$g_i = W_g(x_i \oplus SVD(c_i^1 \oplus \cdots \oplus c_i^6)) \quad (2)$$

where $SVD$ is used to extract key features from commensense, $W_g$ is used to reduce feature dimension. $g_i \in \mathbb{R}^{d_f}$ is the final semantic-view feature of utterance $u_i$.

### 3.2.2 Context-view Feature Extraction

**Correlation Probe**   Given a conversation comprised of $N$ utterances $[u_1, u_2, \cdots, u_N]$, the goal of our probing technique is to derive a function $f(\cdot, \cdot)$ that captures the correaltion between an arbitraty pair of utterances. To this end, we employ a PCM(Gu et al., 2021), which is pretrained on masked utterance generation, next utterance generation and distributed utterance order ranking task.

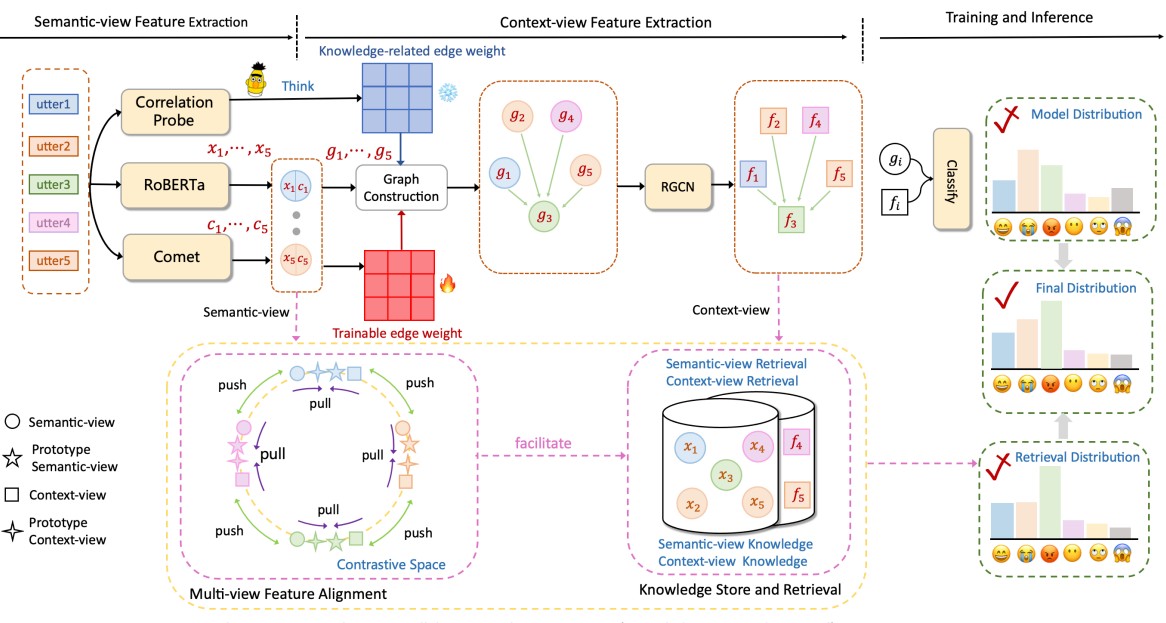

Figure 3: The overall framework of MFAM, which mainly consists of four parts: multi-view feature extraction, multi-view feature alignment, knowledge store and retrieval, training and inference.

As shown in Figure 4, we first feed the $N$ utterances into the PCM. We use $h_j^u$ to denote the contextualized representation of the $j$-th utterance $u_j$, where $1 \leq j \leq N$. Then, we replace the $u_i$ with a mask utterance $[CLS, MASK, SEP]$ and feed the corrupted $N$ utterances into the PCM again. Accordingly, we use $h_{j\backslash i}^u$ to denote the new representation of the $j$-th utterance $u_j$ when $u_i$ is masked out. Formally, we measure the distance between $h_j^u$ and $h_{j\backslash i}^u$ to induce the correlation between $u_i$ and $u_j$ as follows:

$$f(u_i, u_j) = d(h_j^u, h_{j\backslash i}^u) \tag{3}$$

where $d(\cdot, \cdot)$ is the distance metric to measure the difference between two vectors. We use Euclidean distance metric to implement $d(\cdot, \cdot)$:

$$d_{Euc}(u_i, u_j) = \|u_i - u_j\|_2 \tag{4}$$

where $d_{Euc}(\cdot, \cdot)$ denotes a distance function in Euclidean space.

By repeating the above process on each pair of utterances $u_i, u_j$ and calculating $f(u_i, u_j)$, we obtain a correlation matrix $X = \{x_{ij}\}_{i=1,j=1}^{|N|,|N|}$, where $x_{ij}$ denotes the correlation between utterance pair $(u_i, u_j)$.

**Graph Construction** Following Ghosal et al. (2019), a conversation having $N$ utterances is represented as a directed graph $\mathcal{G} = (\mathcal{V}, \mathcal{E}, \mathcal{R}, \mathcal{W})$, with vertices/nodes $v_i \in \mathcal{V}$, labeled edges (relations)

$r_{ij} \in \mathcal{E}$ where $r \in \mathcal{R}$ is the relation type of the edge between $v_i$ and $v_j$ and $\alpha_{ij}$ is the weight of the labeled edge $r_{ij}$, with $0 \leq \alpha_{ij} \leq 1$, where $\alpha_{ij} \in \mathcal{W}$ and $i, j \in [1, 2, ..., N]$.

Each utterance in the conversation is represented as a vertex $v_i \in \mathcal{V}$ in $\mathcal{G}$. Each vertex $v_i$ is initialized with the corresponding semantic-view feature $g_i$, for all $i \in [1, 2, \cdots, N]$, and has an edge with the immediate $p$ utterances of the past: $v_{i-1}, \ldots, v_{i-p}$, $f$ utterances of the future: $v_{i+1}, \ldots, v_{i+f}$ and itself: $v_i$. Considering to speaker dependency and temporal dependency, there exists multiple relation types of edge. The edge weights $W_{ij}$ are obtained by combining the results of two computation ways. One use a similarity based attention module to calculate edge weights $\alpha_{ij}$, weights keep changing during training, which can be regarded as adaptive training. The other takes the correlation $x_{ij}$ between utterance pair $(u_i, u_j)$ computed by Eq.(3) as edge weights, weights keep frozen during training, which can be regarded as correlation-based knowledge injection:

$$\begin{aligned} \alpha_{ij} &= softmax(g_i^T W_e[g_{i-p}, \ldots, g_{i+f}]) \\ W_{ij} &= \alpha_{ij} + bx_{ij} \end{aligned} \tag{5}$$

where $j = i - p, \ldots, i + f$, $b$ denotes the predefined weight coefficient of $x_{ij}$, reflects the injection intensity of correlation-based knowledge.

**Interaction between Utterance** Based on the constructed graph, we utilize RGCN to implement

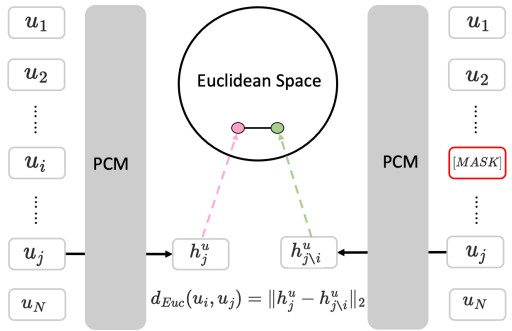

Figure 4: Correlation Probe procedure.

interactions between utterances, and then obtain context-view feature $f_i$ of utterance $u_i$. The detailed calculation process can be found in Appendix A.

## 3.3 Multi-view Feature Alignment

We adopt SCL to align semantic-view and context-view features. Specifically, in a batch consisting of $M$ training samples, for all $m \in [1, 2, ..., M]$, both $g_m$ and $f_m$ are involved in the computation of SCL, we can incorporate $g_m$ and $f_m$ into the SCL computation separately, forming $2M$ samples, or concatenate them for SCL computation. Taking the former approach as example, the supervised contrastive loss of $2M$ samples in a multiview batch can be expressed by the following equation:

$$G_M = [g_1, \ldots, g_m] \quad F_M = [f_1, \ldots, f_m]$$
$$Y = [G_M, \ F_M]$$
$$L_{SCL} = \sum_{i \in I} \frac{-1}{|P(i)|} \sum_{p \in P(i)} \mathrm{SIM}(p, i) \quad (6)$$
$$\mathrm{SIM}(p, i) = \log \frac{\exp((Y_i \cdot Y_p)/\tau)}{\sum_{a \in A(i)} \exp(Y_i \cdot Y_a/\tau)}$$

where $Y \in \mathbb{R}^{2M \times d}$, $i \in I = \{1, 2, \cdots, 2M\}$ indicate the index of the samples in a multiview batch, $\tau \in R^+$ denotes the temperature coefficient used to control the distance between instances, $P(i) = I_{j=i} - \{i\}$ represents samples with the same category as $i$ while excluding itself, $A(i) = I - \{i, i + M/i - M\}$ indicates samples in the multiview batch except itself.

To further enrich the samples under each category, we incorporate the prototype vectors corresponding to each category into the SCL computation. Prototype vectors can correspond to features at different views, forming multiple prototypes, or they can correspond to the concatenation of features at different views, forming single prototype.

For instance, with multiple prototypes, the updates for $P(i)$ and $A(i)$ are as follows:

$$P(i) = I_{j=i} - \{i\} + \{l_c\} + \{o_c\}$$
$$A(i) = I - \{i, i + M/i - M\} + \{\mathcal{L}\} + \{\mathcal{O}\} \quad (7)$$

where $c$ represents the emotion category corresponding to sample $i$, $\mathcal{L} = \{l_1, \cdots, l_{|\mathcal{Y}|}\}$ and $\mathcal{O} = \{o_1, \cdots, o_{|\mathcal{Y}|}\}$ respectively represents the index collection of semantic-view and context-view prototype vectors for all emotion categories.

## 3.4 Knowledge Store and Retrieval

Given a training set $(u, y) \in (\mathcal{U}, \mathcal{Y})$, parametric model computes semantic-view feature $x^2$ and context-view feature $f$ of input utterance $u$. Then we get semantic-view features $\{x_i\}_{i=1}^P$ and context-view features $\{f_i\}_{i=1}^P$ of all training inputs $\{u_i\}_{i=1}^P$. We use BERT-whitening (Su et al., 2021) to reduce the feature dimension, therefore, the retrieval speed is accelerated.

$$\{\widetilde{x}_i\}_{i=1}^P = \mathrm{BERT\text{-}whitening}(\{x_i\}_{i=1}^P)$$
$$\{\widetilde{f}_i\}_{i=1}^P = \mathrm{BERT\text{-}whitening}(\{f_i\}_{i=1}^P) \quad (8)$$

We construct two knowledge stores, one from semantic-view and the other from context-view. Semantic-view knowledge store regarding semantic-view features $\widetilde{x}_i$ and corresponding emotion labels $y_i$ as memorized key-value pairs. For context-view knowledge store, we adopt the same construction method as semantic-view.

$$(\mathcal{K}_{sem}, \mathcal{V}_{sem}) = \bigcup_{(u,y) \in (\mathcal{U}, \mathcal{Y})} (\widetilde{x}_i, y_i)$$
$$(\mathcal{K}_{con}, \mathcal{V}_{con}) = \bigcup_{(u,y) \in (\mathcal{U}, \mathcal{Y})} (\widetilde{f}_i, y_i) \quad (9)$$

During inference, given an utterance $u$, parametric model computes corresponding semantic-view feature $x$ and context-view feature $f$. We do the same as Eq.(8) for $x$ and $f$ to get $\widetilde{x}$ and $\widetilde{f}$, which is used to respectively retrieve the semantic-view and context-view knowledge store for the $k$ nearest neighbors ($k$NN) $\mathcal{U}$ according to L2 distance:

$$p_{kNN-sem}(y|u) \propto \sum_{(k_i, v_i) \in \mathcal{U}} \mathbb{1}_{y=v_i} \mathrm{DIS}(k_i, \widetilde{x})$$
$$p_{kNN-con}(y|u) \propto \sum_{(k_i, v_i) \in \mathcal{U}} \mathbb{1}_{y=v_i} \mathrm{DIS}(k_i, \widetilde{f}) \quad (10)$$

$$\mathrm{DIS}(k_i, \widetilde{x}/\widetilde{f}) = \exp(\frac{-d(k_i, \widetilde{x}/\widetilde{f})}{T})$$

---

[2] Semantic feature is an essential component of the semantic-view features, we use it to represent semantic-view features during knowledge store construction and retrieval.

where $T$ is the temperature.

## 3.5 Training and Inference

**Training**  Since semantic-view and context-view features have been well aligned, we concatenate them for emotion classification and use cross-entropy loss function to calculate classification loss. Meanwhile, we use logit compensation (Menon et al., 2020) to eliminate the bias in the classification layer caused by class imbalance issues:

$$L_{CE} = -\sum_{i=1}^{M} \log \frac{\exp(\varphi_y(w_i) + \delta_y)}{\sum_{y' \in [\mathcal{Y}]} \exp(\varphi_{y'}(w_i) + \delta_{y'})} \quad (11)$$

where $w_i = g_i \oplus f_i$, $\oplus$ represent feature concatention, $\varphi$ represent classification layer, $\delta_y$ is the compensation for class $y$ and its value is related to class-frequency. Finally, we have the following batch-wise loss for training.

$$L = L_{CE} + L_{SCL} \quad (12)$$

**Inference**  During inference, we not only infer the utterance's emotion through the trained model but also assists decision-making by retrieving examples that are memorized in the two knowledge stores:

$$p(y|u) = \lambda p_{Model}(y|u) + \mu p_{kNN-sem}(y|u) \\ + \gamma p_{kNN-con}(y|u) \quad (13)$$

where $\lambda$, $\mu$, $\gamma$ represent the interpolation hyperparameters between model output distribution and $k$NN distribution.

## 4 Experiments

### 4.1 Datasets and Evaluation metrics

We evaluate our method on four benchmark datasets: IEMOCAP (Busso et al., 2008), MELD (Poria et al., 2019a), DailyDialog (Li et al., 2017) and EmoryNLP (Zahiri and Choi, 2018). Detailed descriptions of each dataset can be found in the appendix B. The statistics of four datasets are presented in Table 1.

We utilize only the textual modality of the above datasets for the experiments. For evaluation metrics, we follow (Shen et al., 2021b) and choose micro-averaged F1 excluding the majority class (neutral) for DailyDialog and weighted-average F1 for the other datasets.

| Dataset | | DD | MELD | ENLP | IEMOCAP |
|---|---|---|---|---|---|
| #Dial | Train | 11118 | 1038 | 713 | 120 |
| | Dev | 1000 | 114 | 99 | 120 |
| | Test | 1000 | 280 | 85 | 31 |
| #CLS | | 7 | 7 | 7 | 6 |

Table 1: Statistics of four benchmark datasets.

### 4.2 Implementation Details

The initial weight of RoBERTa come from Huggingface's Transformers(Wolf et al., 2020). We utilize Adam (Kingma and Ba, 2014) optimizer to optimize the network parameters of our model and the learning rate is set to 0.0003 and remains constant during the training process. We adopt faiss (Johnson et al., 2019) library to conduct retrieval, and the dimensions of semantic-view and context-view features respectively become 384 and 64 after the BERT-whitening operation. We search the hyper-parameters on the validation set. All experiments are conducted on A6000 GPU with 48GB memory.

### 4.3 Compared Methods

We compare our model with the following baselines in our experiments.

**Transformer-based methods:**  BERT (Devlin et al., 2018), RoBERTa(Liu et al., 2019), CoG-BART(Li et al., 2021), CoMPM(Lee and Lee, 2021), SPCL(Song et al., 2022), MPLP(Zhang et al., 2023).

**Recurrence-based methods:** DialogueRNN (Majumder et al., 2019), DialogueRNN-RoBERTa, COSMIC (Ghosal et al., 2020), DialogueCRN(Hu et al., 2021), SACL(Hu et al., 2023).

**Graph-based methods:** DialogueGCN (Ghosal et al., 2019), DialogueGCN-RoBERTa, RGAT (Ishiwatari et al., 2020), RGAT-RoBERTa, SGED+DAG-ERC(Bao et al., 2022) and DAG-ERC(Shen et al., 2021b).

In addition to the above-mentioned baselines, we also take into account the performance of Chat-GPT(Brown et al., 2020) on the ERC task evaluated by Yang et al. (2023) and Zhao et al. (2023).

## 5 Results and Analysis

### 5.1 Overall Performance

The comparison results between the proposed MFAM and other baseline methods are reported in Table 2. We can observe from the results that

| Model | IEMOCAP | MELD | DailyDialog | EmoryNLP | Avg |
|---|---|---|---|---|---|
| ChatGPT$^{\dagger}_{zs}$ (Zhao et al., 2023) | 44.97 | 57.30 | 40.66 | 37.47 | 45.10 |
| ChatGPT$^{\ddagger}_{zs}$ (Yang et al., 2023) | 53.35 | 61.18 | 43.27 | 32.64 | 47.61 |
| ChatGPT$^{\dagger}_{fs}$ (Zhao et al., 2023) | 48.58 | 58.35 | 42.39 | 35.92 | 46.31 |
| *Transformer-based Methods* | | | | | |
| BERT (Devlin et al., 2018) | 60.98 | 62.28 | 54.85 | 34.87 | 53.25 |
| RoBERTa (Liu et al., 2019) | 63.38 | 62.51 | 54.33 | 35.90 | 54.03 |
| BART (Lewis et al., 2019) | 56.14 | 63.57 | 55.34 | 35.98 | 52.76 |
| CoG-BART(Li et al., 2021) | 66.18 | 64.81 | 56.29 | 38.04 | 56.33 |
| CoMPM(Lee and Lee, 2021) | 69.46 | 66.52 | 60.34 | 38.93 | 58.81 |
| SPCL(Song et al., 2022)* | 66.93 | 64.93 | - | 39.45 | - |
| MPLP(Zhang et al., 2023) | 66.65 | 66.51 | 59.92 | - | - |
| *Recurrence-based Methods* | | | | | |
| DialogueRNN (Majumder et al., 2019) | 62.75 | 57.03 | - | - | - |
| +RoBERTa* | 64.76 | 63.61 | 57.32 | 37.44 | 55.78 |
| COSMIC (Ghosal et al., 2020) | 63.05 | 64.28 | 56.16 | 37.10 | 55.15 |
| DialogueCRN (Hu et al., 2021) | 66.20 | 58.39 | - | - | - |
| SACL(Hu et al., 2023) | 69.22 | 66.45 | - | 39.65 | - |
| *Graph-based Methods* | | | | | |
| DialogueGCN (Ghosal et al., 2019) | 64.18 | 58.10 | - | - | - |
| +RoBERTa* | 64.91 | 63.02 | 57.52 | 38.10 | 55.89 |
| RGAT (Ishiwatari et al., 2020) | 65.22 | 60.91 | 54.31 | 34.42 | 53.72 |
| + RoBERTa* | 66.36 | 62.80 | 59.02 | 37.89 | 56.52 |
| DAG-ERC (Shen et al., 2021b) | 68.03 | 63.65 | 59.33 | 39.02 | 57.51 |
| SGED + DAG-ERC (Bao et al., 2022) | 68.53 | 65.46 | - | 40.24 | - |
| MFAM(Ours) | **70.16** | **66.65** | **62.19** | **41.06** | **60.02** |

Table 2: Overall results (%) against various baseline methods for ERC on the four benchmarks. † and ‡ represent different prompt templates, $zs$ and $fs$ respectively represent zero-shot and few-shot scenario. * represent models with RoBERTa utterance features. The results reported in the table are from the original paper or their official repository. Best results are highlighted in bold. The improvement of our model over all baselines is statistically significant with $p < 0.05$ under t-test.

our proposed MFAM consistently achieves start-of-the-art results on four widely used benchmarks.

As a graph-based method, MFAM achieves an average performance improvement of +4.36% when compared to preceding graph-based methods. Moreover, when considering the four benchmarks individually, MFAM achieves a performance improvement of +2.38%, +1.82%, +4.82%, and +2.04% respectively, marking a significant enhancement over the earlier graph-based methods.

MFAM also shows a significant advantage when compared to other types of methods. It achieves an average performance gain of +2.06%, +7.60% respectively compared to transformer-based and recurrence-based methods. Moreover, we notice that the performance of ChatGPT in zero-shot and few-shot scenarios still has a significant gap compared to the performance of models currently

trained on the full dataset.

## 5.2 Ablation Study

To study the effectiveness of the modules in MFAM, we evaluate MFAM by removing knowledge module, alignment module and memorization module. The results of ablation study are shown in Table 3.

Concerning the knowledge module, which includes commonsense knowledge and correlation-based knowledge. Its removal results in a sharp performance drop on IEMOCAP and DailyDialog, while a slight drop on MELD and EmoryNLP. Therefore, we can infer that the conversations in IEMOCAP and DailyDialog involve more commonsense knowledge and more complex utterance interactions. Concerning the alignment module, its removal leads to a similar and significant

| Model | IEMOCAP | MELD | DailyDialog | EmoryNLP | Avg |
|---|---|---|---|---|---|
| MFAM(Ours) | 70.16 | 66.65 | 62.19 | 41.06 | 60.02 |
| w/o Know | 68.66(↓ 1.50) | 65.97(↓ 0.68) | 61.04(↓ 1.15) | 40.27(↓ 0.79) | 58.99(↓ 1.03) |
| w/o SCL | 68.64(↓ 1.52) | 65.69(↓ 0.96) | 60.61(↓ 1.58) | 39.74(↓ 1.32) | 58.67(↓ 1.35) |
| w/o Retrieval | 68.46(↓ 1.70) | 64.64(↓ 2.01) | 60.10(↓ 2.09) | 39.10(↓ 1.96) | 58.08(↓ 1.94) |
| w/o Retrieval & SCL | 66.45(↓ 3.71) | 63.46(↓ 3.19) | 58.49(↓ 3.70) | 38.26(↓ 2.80) | 56.67(↓ 3.35) |

Table 3: Results of ablation study on the four benchmrks.

drop in the model's performance on all datasets, demonstrating the importance of multi-view feature alignment in the ERC task. Concerning the memorization module, its removal results in a drastic decrease in model performance across all datasets, highlighting the effectiveness of inferencing through memorization in addressing class imbalance issues. Moreover, the simultaneous removal of the alignment and memorization modules results in a performance decline that is greater than the sum of the declines caused by the removal of each module individually, proving that aligning semantic-view and context-view features can facilitate the joint retrieval of the semantic-view and context-view.

### 5.3 Analysis on $\lambda, \mu, \gamma$

$\lambda$, $\mu$ and $\gamma$ are very important parameters, which respectively represent the weights occupied by the model, semantic-view retrieval, and context-view retrieval during inference. Determining the appropriate weight combination to reinforce their interplay is very important. Table 4 shows the test f1-scores on the IEMOCAP dataset for different $\lambda$, $\mu$ and $\gamma$.

We can observe that when setting $\mu$ to 0.2, dynamically adjusting $\lambda$ and $\gamma$ leads to continuous changes in performance, when the value of $\lambda$ rises to 0.7, corresponding to a drop in $\gamma$ to 0.1,

| $\lambda$ | $\mu$ | $\gamma$ | Performance |
|---|---|---|---|
| 0.4 | 0.2 | 0.4 | 67.67 |
| 0.5 | 0.2 | 0.3 | 67.48 |
| 0.6 | 0.2 | 0.2 | 68.91 |
| 0.65 | 0.2 | 0.15 | 69.26 |
| 0.7 | 0.15 | 0.15 | 68.76 |
| **0.7** | **0.2** | **0.1** | **70.16** |
| 0.7 | 0.25 | 0.05 | 68.98 |
| 0.75 | 0.2 | 0.05 | 69.02 |

Table 4: Test f1-scores on IEMOCAP dataset with different $\lambda$, $\mu$ and $\gamma$.

the best performance is achieved. Continuing to increase the value of $\lambda$ and reduce the value of $\gamma$ would result in a performance decline. In addition, fluctuating the value of $\mu$ while keeping $\lambda$ and $\gamma$ at 0.7 and 0.1 respectively also leads to a decrease in performance.

### 5.4 Visualization on Multi-view Feature

To conduct a qualitative analysis of multi-view feature alignment, we utilize t-sne(Van der Maaten and Hinton, 2008) to visualize the prototype vectors corresponding to semantic-view and context-view features under each emotion category, as shown in Figure 5.

It can be observed that semantic-view and context-view features under the same emotion category are well aligned. Meanwhile, positive emotions such as "happy" and "excited" are close to each other, while negative emotions like "frustrated", "angry", and "sad" are also close to each other.

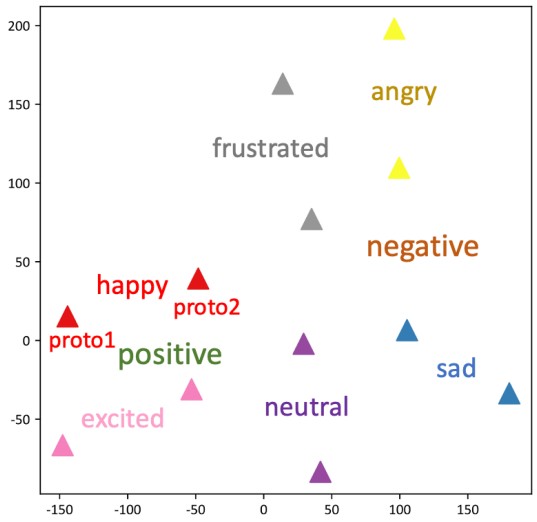

Figure 5: Visualize the prototype vectors corresponding to semantic-view and context-view features under each emotion category. Proto1 and proto2 correspond to semantic-view and context-view features, respectively.

# 6 Conclusion

In this paper, we propose **M**ulti-view **F**eature **A**lignment and **M**emorization (**MFAM**) for ERC. Firstly, in order to obtain accurate context-view features, we treat the PCM as a prior knowledge base and from which we elicit correlations between utterances by a probing procedure. Then we adopt SCL to align semantic-view and context-view features. Moreover, we improve the recognition performance of tail-class samples by retrieving examples that are memorized in the two knowledge stores during inference. We achieve state-of-the-art results on four widely used benchmarks, ablation study and visualized results demonstrates the effectiveness of multi-view feature alignment and memorization.

## Limitations

There are two major limitations in this study. Firstly, semantic-view and context-view retrieval based on the training set may suffer from dataset and model bias. Secondly, during inference, we need to consider three probability distributions: semantic-view retrieval, context-view retrieval, and model output. Determining the appropriate weight combination to reinforce their interplay is very important, therefore, additional computational resources are required to find these parameters. The aforementioned limitations will be left for future research.

## Acknowledgement

This work is supported by the Fundamental Research Funds for the Central Universities (No. 226-2022-00143), National Key Research and Development Project of China (No. 2018AAA0101900), and MOE Engineering Research Center of Digital Library.

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

## A  Interaction between Utterance

Following Ghosal et al. (2019), we use a two-step graph convolution process to obtain context-view features. In the first step, a new feature vector $h_i^{(1)}$ is computed for vertex $v_i$ by aggregating local neighbourhood information using the relation specific transformation inspired from (Schlichtkrull et al., 2018):

$$h_i^{(1)} = \sigma(\sum_{r \in \mathcal{R}} \sum_{j \in N_i^r} \frac{W_{ij}}{c_{i,r}} W_r^{(1)} g_j + W_{ii} W_0^{(1)} g_i)$$

$$\text{for } i = 1, 2, \ldots, N \tag{14}$$

where $W_{ij}$ and $W_{ii}$ are the edge weights, $N_i^r$ denotes the neighbouring indices of vertex $i$ under relation $r \in \mathcal{R}$, $c_{i,r} = |N_i^r|$. $\sigma$ is an activation function such as ReLU, $W_r^{(1)}$ and $W_0^{(1)}$ are learnable parameters of the transformation. In the second step, another local neighbourhood based transformation is applied over the output of the first step:

$$f_i = \sigma(\sum_{j \in N_i^r} W^{(2)} h_j^{(1)} + W_0^{(2)} h_i^{(1)})$$

$$\text{for } i = 1, 2, \ldots, N \tag{15}$$

where, $W^{(2)}$ and $W_0^{(2)}$ are parameters of these transformation and $\sigma$ is the activation function.

## B  Detailed Descriptions of ERC Datasets

**IEMOCAP** (Busso et al., 2008): Multimodal ERC dataset. Each conversation within the IEMOCAP dataset comes from the performance based on script by two actors. Models are evaluated on the samples with 6 types of emotion, namely *neutral*, *happiness*, *sadness*, *anger*, *frustrated*, and *excited*.

**MELD** (Poria et al., 2019a): Multimodal ERC dataset gathered from the TV show *Friends*. 7 emotion labels are included: *neutral*, *happiness*, *surprise*, *sadness*, *anger*, *disgust*, and *fear*.

**DailyDialog** (Li et al., 2017): Dialogues penned by humans, collected from communications of English learners. 7 emotion labels are included: *neutral*, *happiness*, *surprise*, *sadness*, *anger*, *disgust*, and *fear*.

**EmoryNLP** (Zahiri and Choi, 2018): This dataset comprises TV show scripts obtained from the television series "Friends," with variations in scene selection and emotion labeling. 7 emotion labels are included: *neutral*, *sad*, *mad*, *scared*, *powerful*, *peaceful*, and *joyful*.

## C  Experimental Results and Analysis

### C.1  Generalization ability of the model

Based on the IEMOCAP dataset, we applied our proposed method to two graph network models, RGAT and DAG-ERC, and two non-graph network models, RoBERTa and COG-BART. The experimental results are shown in table 5.

| Model | Original | Original+Ours |
|---|---|---|
| RGAT | 65.22 | 69.35 |
| DAG-ERC | 68.03 | 70.27 |
| RoBERTa | 63.38 | 65.96 |
| COG-BART | 66.18 | 69.12 |

Table 5: Generalization ability of the proposed model. For the RoBERTa model, since it does not model the context-view features, we only apply the memorization technique.

Based on the experimental results, applying our method to other graph networks and non-graph structured models can bring significant performance improvements, indicating that our proposed method has good generalization capabilities.

### C.2  Case Study

After eliciting correlations between utterances by a probing procedure from a pre-trained conversation model, the utterance will pay attention to the coherence of the context and the overall logic flow of the dialogue. Here is a case in table 6, where $u_3$ nicely picks up the emotion of $u_1$, and the emotions of $u_{17}$ and $u_{18}$ also correspond well. Meanwhile, by examining the relevance scores between $u_{17}$ and other utterances, it is found that $u_{17}$ has noticed the topic change at $u_6$ (a more sorrowful topic turning into an entertaining one).

| Index | Conversational Context | w/o probe | with probe |
|---|---|---|---|
| $u_1$ | B: I'm really fed up with work at the moment . I need a break! | Disgust | Disgust |
| $u_2$ | A: are you doing anything this weekend ? | Neutral | Neutral |
| $u_3$ | B: I have to work on Saturday all day ! I really hate my job ! | Anger | Disgust |
| $u_4$ | A: are you available on Sunday ? | Neutral | Neutral |
| $u_5$ | B: yes , that's my only day off until Thursday. | Neutral | Neutral |
| $u_6$ | A: My friends and I are planning on going to the beach. Come with us ? | Neutral | Neutral |
| $u_{17}$ | B: ok , where and when should I meet you ? | Neutral | Happiness |
| $u_{18}$ | A: we'll pick you up at your place at noon . Be there or be square ! | Happiness | Happiness |

Table 6: A case that our method gives the correct result.