# OpenReview forum: "Enhancing Emotion Recognition in Conversation via Multi-view Feature Alignment and Memorization"
_EMNLP/2023/Conference — EMNLP 2023 Findings_

### Official Review · Reviewer_oCgX · 2023-07-31

**Typos Grammar Style And Presentation Improvements:** See Questions for Authors.
**Soundness:** 3

**Excitement:**

3: Ambivalent: It has merits (e.g., it reports state-of-the-art results, the idea is nice), but there are key weaknesses (e.g., it describes incremental work), and it can significantly benefit from another round of revision. However, I won't object to accepting it if my co-reviewers champion it.

**Missing References:**

NA.

**Paper Topic And Main Contributions:**

The manuscript proposes a multi-view feature alignment and memorization framework for emotion recognition in conversation. The main contributions are aligning semantic-view and context-view features, memorization techniques for improving tail-class classification, modeling the interactions between utterances via probing, and state-of-the-art results on four benchmarks.

**Questions For The Authors:**

1. What is "PLMs" in the abstract? Assuming all readers are familiar with certain abbreviations can lead to poor readability.
2. In lines 039-041, the definition of ERC is to predict emotion labels of each utterance based on the surrounding context, which seems a bit confusing. Even when not in a conversation, I think this definition is still valid, then why is it called emotion recognition in conversation?
3. Conflicting claims: In lines 056-060, the authors analyzed that it's difficult to fully model interaction simply through a graph neural network, but the proposed MFAM is based on graph neural networks.
4. How is Figure 2 obtained? What are the exact meanings of points in different colors? How do you judge that they are context-view or semantic-view? Giving a figure of two separate parts of points cannot convince the audience.
5. Why do the authors extract commonsense features in Section 3.2.1?
6. Any theoretical reason for choosing Euclidean distance in equation (4)?
7. Equation (3) uses $f$ to denote correlation, then in line 266, $f$ denotes future utterances, then in line 286, $f$ denotes the context-view feature; This leaves a bad impression and the authors should be careful when writing notations.
8. How is the determination of b in Equation (5)?
9. In lines 283-288, I understand the space is limited, but at least here the basic process should be explained clearly instead of referring readers to the appendix. What is the two-step graph convolution process? Assuming all readers know how GCN works is already not appropriate, leave alone a "two-step" GCN which may contain some new operations.
10. Any specific reason for choosing BERT-whitening models for dimension reduction in Equation (8)?

**Reasons To Accept:**

1. State-of-the-art results on four benchmarks and extensive experiments.
2. An interesting manner of calculating correlations between utterances, while the novelty is not guaranteed.

**Reasons To Reject:**

1. Relatively weak or unclear motivation: the features of semantic-view and context-view are not aligned, but why should they be aligned? What's the damage if they are not? Why is it difficult to fully model interactions through a graph neural network? More insightful motivations should be given.
2. The contribution of the memorization technique is relatively invalid since the proposed technique is basically storing pairs of feature vectors and labels and using clustering techniques to help classification. This is a cliche technique, but the authors use new models within.
3. Poor readability with many confusing descriptions, see Questions for Authors for details.

**Reproducibility:**

4: Could mostly reproduce the results, but there may be some variation because of sample variance or minor variations in their interpretation of the protocol or method.

**Reviewer Confidence:**

3: Pretty sure, but there's a chance I missed something. Although I have a good feel for this area in general, I did not carefully check the paper's details, e.g., the math, experimental design, or novelty.

---

> ### Author Rebuttal · Authors · 2023-08-29
>
> Thank you for your valuable time and reviews.
>
> **Response to R1**: *about the motivation of the paper
>
> This paper first presents several major challenges existing in ERC tasks, including complex emotion interaction between utterances, class imbalance and fine-grained classification problems.
>
> For addressing the challenge of complex emotional interactions between utterances, we use a pre-trained conversation model to implement the injection of the dialogue system's prior knowledge to fully models the interactions between utterances. However, previous methods solely use graph neural networks for modeling.
>
> Meanwhile, We discover the phenomenon of feature separation between semantic-view and context-view. Other methods may consider using residual connections to connect the features of these two views. However, as shown in Figure 2 in the paper, the features of the two views are separated, and directly performing residual connections will lead to bias in the representation. Considering that some utterances need to categorize emotions more from the semantic-view, then the context-view needs to move closer to the semantic-view, and vice versa. In addition, both views are important for emotion recognition and represent the expression of emotions in different views. Therefore, we propose for the first time to align the features of the semantic-view and context-view to form a unified emotional space. This unified emotional space provides multiple views for recognizing the emotions of utterances.
>
> For addressing the long-tail challenge, we apply the memorization technique to the ERC task for the first time.
>
> **It is worth noting that the concept of a unified emotional space effectively connects the methods proposed in this paper**. By fully modeling the features of context-view, a better emotional space is formed. The memorization technique is also based on the unified emotional space to achieve joint retrieval of semantic-view and context-view features.
>
> *about the alignment
>
> Some utterances need to categorize emotions more from the semantic-view, then the context-view needs to move closer to the semantic-view, and vice versa. Both views are important for emotion recognition and represent the expression of emotions in different views. Therefore, we consider aligning the features of the semantic-view and context-view to form a unified emotional space.
> *about the interaction between utterances
>
> If we do not align the features of these two views and directly use context-view or semantic-view for emotion recognition, the information of one view will be lost. If we directly perform residual connections between the features of the two views, it will cause a bias in the representation since they are separate from each other.
>
> *about the interaction between utterances
>
> Intra-speaker and inter-speaker interactions between utterances mimic the emotional inertia and emotional stimulus, while global and local interactions represent the overall and specific emotional development in conversations. Previous methods often involved setting a fixed window size and conducting interactions between utterances within the window. However, in order to fully model the interactions between utterances, relying solely on graph neural networks is not sufficient, prior knowledge of the dialogue system itself is also needed. We believe that pre-trained conversational models have the ability to grasp interactive relationships that graph neural networks are unable to learn. Ablation experiments have demonstrated the efficiency of incorporating prior knowledge of the dialogue system itself.
>
> **Response to R2**: Firstly, to the best of our knowledge, we are the first to apply memorization to ERC tasks. Secondly, memorization techniques do not exist in isolation, but are built upon a unified emotional space. Based on the unified emotional space, which aligns semantic-view and context-view features, memorization performs joint semantic-view and context-view features retrieval.
>
> **Response to R3, Q1**: Thank you for your suggestion. The PLMs mentioned here are pre-trained language models. In the future, we will pay attention to the readability issues you mentioned.
>
> **Response to R3, Q2**: We highly agree with your opinion, and when defining it here, we need to incorporate the element of dialogue. For example, ERC is to assign emotion labels to all the utterances within a dialogue from a pre-deﬁned emotion category set.
>
> **Response to R3, Q3**: Apologies for causing any confusion. Let us explain to you that simply using graph neural networks for interaction between utterances is not enough, so we injected the prior knowledge of the dialogue system itself during the calculation process of the graph neural network.
>
> **Response to R3, Q4**: After the model training is close to convergence, the dataset samples are passed through the model to obtain the features of their semantic-view and context-view, and visualized with the help of t-SNE technology. When drawing the figure, we first draw the feature distribution of the semantic-view, then the feature distribution of the context-view, and finally summarize them in one figure. Therefore, we can find that there is a bias in the feature representation.
>
> **Response to R3, Q5**: Extracting utterance-related commonsense features helps to understand the speaker's intention and behavior, such as xWant representing what the speaker may want to do after the event, and xIntent representing the reason why the speaker would cause the event. Moreover, commonsense features also help in understanding colloquial expressions in dialogues.
>
> **Response to R3, Q6**: Since Euclidean space is a relatively common type of metric space and can effectively measure the distance between vectors, it was initially considered. Inspired by the question you raised, we researched other metric spaces and found that non-linear spaces, such as hyperbolic spaces, can better model the non-linear and hierarchical dependencies between utterances. We conducted preliminary experiments on MELD and found that hyperbolic space might have advantages over Euclidean space in modeling the correlations between utterances.
> | Space            | Performance |
> | ---------------- | ----------- |
> | Euclidean space  | 66.65       |
> | Hyperbolic Space | 66.89       |
>
>
> **Response to R3, Q7**: Thank you for your suggestion. We will pay special attention to the standardized use of symbols in the future.
>
> **Response to R3, Q8**: b represents the injection intensity of prior knowledge related to the dialogue system, so we consider its value should not be too large. We treat b as a hyperparameter, and traverse it within the range of 0.05-0.20 with a step size of 0.05, choosing the parameter with better performance. For example, in the IEMOCAP dataset, a good performance is achieved when b is 0.1.
>
> **Response to R3, Q9**: Thank you for your suggestion. We will incorporate the basic process into the main text of the paper. The first step of the operation is to compute a new feature for each utterance by aggregating local neighborhood information using the relation speciﬁc transformation, which takes into account different types of relationships such as intra-speaker, inter-speaker, past utterances, and future utterances. In the second step, another local neighborhood-based transformation is applied over the output of the ﬁrst step.
>
> **Response to R3, Q10**: BERT-whitening employs the SVD algorithm to achieve dimension reduction while enhancing the isotropy of sentence representations. Effectively improve retrieval results and accelerate retrieval speed.

---

### Official Review · Reviewer_Gm2P · 2023-08-03

**Soundness:** 3

**Excitement:**

3: Ambivalent: It has merits (e.g., it reports state-of-the-art results, the idea is nice), but there are key weaknesses (e.g., it describes incremental work), and it can significantly benefit from another round of revision. However, I won't object to accepting it if my co-reviewers champion it.

**Paper Topic And Main Contributions:**

This paper proposes a new paradigm, treating a pre-trained conversation model as a knowledge base, eliciting correlations between utterances through probing, and using supervised contrastive learning to align semantic and context features. Additionally, they introduce semi-parametric inferencing through memorization to handle tail class recognition. Experimental results demonstrate that the proposed approach achieves state-of-the-art results on four benchmarks.

**Questions For The Authors:**

Question A： The abstract looks a little vague. For example, “However, it is difficult to fully model interaction between utterances …” What is 'interaction between utterances' and why is it difficult to model? This information is not evident from the previous context. Additionally, the misalignment between the two views might seem obvious since most ERC models aggregate information using methods like residual connections. So, why the need for alignment? Isn't the goal to leverage the advantages of both features? Or does alignment help in achieving a balance between both features?
Question B： How well does the proposed method generalize, and can it achieve improvements in other graph networks or non-graph-structured models?
Question C： After addressing the issue of tail class sample recognition, does the model show improvement on samples with few classes? Does the performance on samples with multiple classes get affected?

I'm happy to update the review if the authors can answer my concerns.

**Reasons To Accept:**

-	Reasonable method.
-	The proposed method outperforms the baselines.

**Reasons To Reject:**

-	Very difficult to follow the motivation of this paper. And it looks like an incremental engineering paper.
-	The abstract looks a little vague. For example, “However, it is difficult to fully model interaction between utterances …” What is 'interaction between utterances' and why is it difficult to model? This information is not evident from the previous context. Additionally, the misalignment between the two views might seem obvious since most ERC models aggregate information using methods like residual connections. So, why the need for alignment? Isn't the goal to leverage the advantages of both features? Or does alignment help in achieving a balance between both features?
-	The author used various techniques to enhance performance, including contrastive learning, external knowledge, and graph networks. However, these methods seem contradictory to the limited experiments conducted. For example, the author proposes a new semi-parametric inferencing paradigm involving memorization to address the recognition problem of tail class samples. However, the term "semi-parametric" is not clearly defined, and there is a lack of experimental evidence to support the effectiveness of the proposed method in tackling the tail class samples problem.
-	The Related Work section lacks a review of self-supervised contrastive learning in ERC.
-	The most recent comparative method is still the preprint version available on ArXiv, which lacks convincing evidence.
-	Table 3 needs some significance tests to further verify the assumptions put forward in the paper.
-	In Chapter 5.3, the significant impact of subtle hyperparameter fluctuations on performance raises concerns about the method's robustness. The authors could consider designing an automated hyperparameter search mechanism or decoupling dependencies on these hyperparameters to address this.
-	There is a lack of error analysis.
-	The formatting of the reference list is disorganized and needs to be adjusted.
-	Writing errors are common across the overall paper.

**Reproducibility:**

3: Could reproduce the results with some difficulty. The settings of parameters are underspecified or subjectively determined; the training/evaluation data are not widely available.

**Reviewer Confidence:**

4: Quite sure. I tried to check the important points carefully. It's unlikely, though conceivable, that I missed something that should affect my ratings.

---

> ### Author Rebuttal · Authors · 2023-08-29
>
> Thank you for your valuable time and reviews.
>
> **Response to R1**: *about the motivation of the paper
>
> This paper first presents several major challenges existing in ERC tasks, including complex emotion interaction between utterances, class imbalance and fine-grained classification problems.
>
> For addressing the challenge of complex emotional interactions between utterances, we use a pre-trained conversation model to implement the injection of the dialogue system's prior knowledge to fully models the interactions between utterances. However, previous methods solely use graph neural networks for modeling.
>
> Meanwhile, We discover the phenomenon of feature separation between semantic-view and context-view. Other methods may consider using residual connections to connect the features of these two views. However, as shown in Figure 2 in the paper, the features of the two views are separated, and directly performing residual connections will lead to bias in the representation. Considering that some utterances need to categorize emotions more from the semantic-view, then the context-view needs to move closer to the semantic-view, and vice versa. In addition, both views are important for emotion recognition and represent the expression of emotions in different views. Therefore, we propose for the first time to align the features of the semantic-view and context-view to form a unified emotional space. This unified emotional space provides multiple views for recognizing the emotions of utterances.
>
> For addressing the long-tail challenge, we apply the memorization technique to the ERC task for the first time.
>
> **It is worth noting that the concept of a unified emotional space effectively connects the methods proposed in this paper**. By fully modeling the features of context-view, a better emotional space is formed. The memorization technique is also based on the unified emotional space to achieve joint retrieval of semantic-view and context-view features.
>
> **Response to R2, Q1**: *about the interaction between utterances
>
> During a conversation, the emotion of utterance expressed by a speaker is influenced by the other speaker's utterances and their own emotional state over the utterances, utilizing relational graph convolution networks to implement interactions between utterances can take into account emotional stimulus from other speakers and their own emotional inertia. Previous methods often involved setting a fixed window size and conducting interactions between utterances within the window. However, in order to fully model the interactions between utterances, relying solely on graph neural networks is not sufficient, prior knowledge of the dialogue system itself is also needed. We believe that pre-trained conversational models have the ability to grasp interactive relationships that graph neural networks are unable to learn. Ablation experiments have demonstrated the efficiency of incorporating prior knowledge of the dialogue system itself.
>
> *about the alignment
>
> Most ERC models aggregate information using methods like residual connections, However, as shown in Figure 2 in the paper, the features of the two views are separated, and directly performing residual connections will lead to bias in the representation. Our motivation is that the semantic-view and context-view features should be unified in the same emotional space, rather than being separate. Some utterances need to categorize emotions more from the semantic-view, then the context-view needs to move closer to the semantic-view, and vice versa. Both views are important for emotion recognition and represent the expression of emotions in different views. As shown in Figure 5 of the paper, the features of the two views come close together to form a unified emotional space.
>
>
> **Response to R3, Q3**: In the ablation study section, we demonstrated the effectiveness of the proposed modules. The term "semi-parametric"  refers to the fact that the model's parameters are obtained through training, while the memorization technique is parameter-free. Therefore, the combination of the two is considered semi-parametric.
>
> For performance of the model on multiple classes and tail classes, as shown in the table below, after introducing the memorization technique, the model has significantly improved recognition performance on tail-class samples. In addition, the model also improved the performance for almost all of the multiple-class samples.
>
> | Emotion class | Neu   | Sur   | Fea*  | Sad   | Joy   | Dis   | Ang   |
> | ------------- | ----- | ----- | ----- | ----- | ----- | ----- | ----- |
> | **MELD**          | +0.75 | +1.79 | +5.49 | +2.11 | +1.09 | +1.35 | +1.76 |
> | Emotion class | Hap*  | Sad   | Neu   | Ang   | Exc   | Fru   | -     |
> | **IEMOCAP**       | +3.12 | +0.96 | +0.89 | +1.43 | +2.31 | +0.59 |       |
> | Emotion class | Joy   | Mad   | Pea   | Neu   | Sad*  | Pow   | Sca   |
> | **EmoryNLP**      | +1.88 | +3.75 | +2.58 | -0.06 | +1.89 | +2.87 | +1.29 |
> | Emotion class | Ang   | Dis   | Fea*  | Hap   | Sad   | Sur   | Neu   |
> | **DailyDialog**   | +3.12 | +2.89 | +3.45 | +1.45 | -0.23 | +0.66 | -     |
>
> Note: * represents tail class.
>
> Analysis: In the final recognition process, memorization and model work together for collaborative inference. For the multiple-class samples, the model's decision results have a higher level of confidence, so the decision results of memorization may not affect the final recognition result. For the tail-class samples, the confidence of the model's decision is relatively low, so the memorization's decision results may affect the final recognition results. The formula for collaborative inference is described as:
> $$p_{final} = \lambda p_{model} + (1 - \lambda) p_{memorization}$$
> where $\lambda$ = 0.7 for the IEMOCAP dataset, the collaborative inference examples of multiple-class and tail-class are shown in the table below.
>
>
>
> | $p_{model}$                      | $p_{memorization}$                | $p_{final}$                              | recognition result |
> | ------------------------------ | ------------------------------- | ------------------------------------- | ------------------ |
> | [**0.9**,0.02,0.02,0.03,0.01,0.02] | [0.25,**0.35**,0.15,0.15,0.05,0.05] | [**0.705**,0.119,0.059,0.066,0.022,0.029] | class 0            |
> | [0.35,**0.4**,0.05,0.05,0.05,0.1]  | [**0.5**,0.1,0.1,0.1,0.1,0.1]       | [**0.395**,0.31,0.065,0.065,0.065,0.1]    | class 0            |
>
>
> **Response to R4**:  Since the method in this paper is based on supervised contrastive learning, the self-supervised part has not been described. We will supplement it accordingly. Thank you for your suggestion.
>
> **Response to R5**: In the results comparison section, we compared various methods. At the same time, the paper you mentioned is the latest research result accepted by ACL 2023.  The data in the Arxiv version is consistent with the ACL 2023 version, and in our revised version, we will change the citation to be from ACL.
>
> **Response to R6**: We run 10 times on each dataset, and based on the results of these 10 runs, we perform a one-sample t-test. The test results are shown in the table below. From the experimental results, it can be seen that the improvement of our model over all baselines is statistically signiﬁcant with p<0.05.
>
> | Dataset     | p     | t     | df   |
> | ----------- | ----- | ----- | ---- |
> | IEMOCAP     | 0.018 | 2.873 | 9    |
> | MELD        | 0.045 | 2.320 | 9    |
> | DailyDialog | 0.001 | 5.072 | 9    |
> | EmoryNLP    | 0.025 | 2.693 | 9    |
>
> **Response to R7**: Thank you for your suggestion, we will learn from your thoughts in the future. Your concern about robustness may be due to the large step size we used when traversing hyperparameters. We will consider using AutoML to automatically search for the best hyperparameters in the future.
>
> **Response to R8**: Thank you for your suggestion, we will add the figure of error analysis to the paper.
>
> **Response to R9-R10**: Thank you for your suggestion, we will supplement and modify the content of the paper according to your reviews.
>
> **Response to Q2**: Based on the IEMOCAP dataset, we applied our proposed method to two graph network models, RGAT and  DAG-ERC, and two non-graph network models,  Roberta and COG-BART. The experimental results are shown in the following table.
>
> | Model    | Original | Original + Our proposed method |
> | -------- | -------- | ------------------------------ |
> | RGAT     | 65.22    | 69.35（+4.12）                 |
> | DAG-ERC  | 68.03    | 70.27（+2.24）                 |
> | Roberta  | 63.38    | 65.96（+2.58）                 |
> | COG-BART | 66.18    | 69.12（+2.94）                 |
>
> Note: For the Roberta model, since it does not model the context-view features, we only apply the memorization technique.
>
> Analysis: Based on the experimental results, applying our method to other graph networks and non-graph structured models can bring significant performance improvements, indicating that our proposed method has good generalization capabilities.

---

### Official Review · Reviewer_NUwX · 2023-08-31

**Soundness:** 4

**Excitement:**

4: Strong: This paper deepens the understanding of some phenomenon or lowers the barriers to an existing research direction.

**Paper Topic And Main Contributions:**

The paper addresses the problem of emotion recognition in conversation where the goal is to identify the emotions conveyed in utterances within the dialogue. The paper proposes Multi-view Feature Alignment and Memorization (MFAM) method which model weights in GNN by eliciting correlations between utterances by a probing procedure from a pre-trained conversation model. The paper also adopts a supervised contrastive learning to align the features at semantic-view and context-view. Furthermore, the paper addresses the problem of tail class recognition by proposing a new semi-parametric paradigm of inference through memorization. Experiments demonstrate the utility of the proposed approach.

**Reasons To Accept:**

The paper is well-motivated and well-written. It addresses an important problem in dialog and the proposed multi-view feature alignment and memorization is interesting and has potential to be used across other tasks. Extensive experiments and comparison with several baselines sufficiently demonstrate the effectiveness of the proposed approach.

**Reasons To Reject:**

I do not find any major weaknesses with the proposed work. Although some of the sub-sections (in section 3) took me multiple readings, the figures helped me to follow the key idea. One aspect that I feel is missing in the current version is the qualitative analysis (I couldn’t find them in supplementary material as well). It would be very helpful to look into some of the detailed qualitative examples/results (I see some analysis in section 5.4).

**Reproducibility:**

4: Could mostly reproduce the results, but there may be some variation because of sample variance or minor variations in their interpretation of the protocol or method.

**Reviewer Confidence:**

3: Pretty sure, but there's a chance I missed something. Although I have a good feel for this area in general, I did not carefully check the paper's details, e.g., the math, experimental design, or novelty.

**Typos Grammar Style And Presentation Improvements:**

The paper has few typos (example: Visulization -> Visualization in line 500). Please correct them in final version.

---

### Meta-Review · Area_Chair_k5CS · 2023-10-05

**Recommendation:** 3

**Metareview:**

This paper describes a new approach to emotion recognition in context by combining a number of techniques. Reviewers were all impressed by the performance of the model, though there were multiple requests for additional comparisons/experiments to understand the performance more (e.g., through an error analysis). The approach represented a meaningful engineering contribution to ERC. However, reviewers were mixed on understanding the motivation for the approach itself and whether the individual pieces (e.g., the graph neural network) were contributing or interacting to solve the ERC task. Some of these issues were caused by the clarity in the writing, though one reviewer was able to follow completely. Overall, found the paper's methods promising but highlighted multiple ways the manuscript could be improved to strengthen the paper's clarity and subsequent impact.

---

### Decision · Program_Chairs · 2023-10-07

**Decision:**

Accept-Findings

**Comment:**

This paper describes a new approach to emotion recognition in context by combining a number of techniques. Reviewers were all impressed by the performance of the model, though there were multiple requests for additional comparisons/experiments to understand the performance more (e.g., through an error analysis). The approach represented a meaningful engineering contribution to ERC. However, reviewers were mixed on understanding the motivation for the approach itself and whether the individual pieces (e.g., the graph neural network) were contributing or interacting to solve the ERC task. Some of these issues were caused by the clarity in the writing, though one reviewer was able to follow completely. Overall, found the paper's methods promising but highlighted multiple ways the manuscript could be improved to strengthen the paper's clarity and subsequent impact.